# Investigations by AFM of Ageing Mechanisms in PLA-Flax Fibre Composites during Garden Composting

**DOI:** 10.3390/polym13142225

**Published:** 2021-07-06

**Authors:** Alessia Melelli, Delphin Pantaloni, Eric Balnois, Olivier Arnould, Frédéric Jamme, Christophe Baley, Johnny Beaugrand, Darshil U. Shah, Alain Bourmaud

**Affiliations:** 1Univ. Bretagne Sud, UMR CNRS 6027, IRDL, 56100 Lorient, France; alessia.melelli@univ-ubs.fr (A.M.); delphin.pantaloni@univ-ubs.fr (D.P.); christophe.baley@univ-ubs.fr (C.B.); 2Univ. Brest, EA3884, LBCM, IUEM, 56100 Lorient, France; eric.balnois@univ-brest.fr; 3LMGC, Université de Montpellier, UMR CNRS 5508, 34090 Montpellier, France; olivier.arnould@umontpellier.fr; 4Synchrotron SOLEIL, DISCO Beamline, 91190 Gif-sur-Yvette, France; frederic.jamme@Synchrotron-Soleil.fr; 5UR1268 Biopolymères Interactions Assemblages, INRAE, 44000 Nantes, France; johnny.beaugrand@inrae.fr; 6Centre for Natural Material Innovation, Department of Architecture, University of Cambridge, Cambridge CB2 1PX, UK; dus20@cam.ac.uk

**Keywords:** AFM, flax fibre, PLA, structural degradation, mechanical properties, SEM

## Abstract

PLA-flax non-woven composites are promising materials, coupling high performance and possible degradation at their end of life. To explore their ageing mechanisms during garden composting, microstructural investigations were carried out through scanning electron microscopy (SEM) and atomic force microscopy (AFM). We observe that flax fibres preferentially degrade ‘inwards’ from the edge to the core of the composite. In addition, progressive erosion of the cell walls occurs within the fibres themselves, ‘outwards’ from the central lumen to the periphery primary wall. This preferential degradation is reflected in the decrease in indentation modulus from around 23 GPa for fibres located in the preserved core of the composite to 3–4 GPa for the remaining outer-most cell wall crowns located at the edge of the sample that is in contact with the compost. Ageing of the PLA matrix is less drastic with a relatively stable indentation modulus. Nevertheless, a change in the PLA morphology, a significant decrease in its roughness and increase of porosity, can be observed towards the edge of the sample, in comparison to the core. This work highlights the important role of intrinsic fibre porosity, called lumen, which is suspected to be a major variable of the compost ageing process, providing pathways of entry for moisture and microorganisms that are involved in cell wall degradation.

## 1. Introduction

Our collective need for sustainable development, driven by environmental regulations, encourages innovation in materials. Flax fibres, thanks to their specific mechanical properties, enable the design of functional, eco-friendly composite materials [1]. To design fully biodegradable or compostable composites, flax fibres must be embedded in biodegradable matrices such as poly-(butylene-succinate) (PBS), poly-(hydroxy-alkanoates) (PHA) or poly-(lactide) (PLA) [2]. In this case, an extended range of end-of-life routes can be planned, avoiding incineration, notably by recycling and by composting.

In the present work, we focus on non-woven flax-PLA composites, due to their availability and the general interest of industry in these two bio-based components, especially for short life applications or when degradation at end of life is expected, such as specific parts for the advertising sector. During composting, degradation mechanisms of the composite material are naturally impacted by environmental conditions but also by the composite structure and the nature (physical, biochemical, thermal) of the polymer and fibre. Composting is a bioprocess involving microorganisms and their colonisation capacities, as well as their arsenal of enzymatic degradation. The polymeric chains of materials can be split and, at the macroscale, the composites will undergo macro- and micro-fragmentation. In compost conditions, PLA degradation is strongly affected by temperature, which substantially impacts the extent and rate of decrease in composite mass and molecular weight [3,4]. Morphologically speaking, degradation induces macro- and micro-fragmentation of the polymer matrix and, when plant fibres are incorporated, structural degradation is often boosted due to the fibres’ hydrophilicity, leading to extensive interfacial damage [2]. At the composite scale, the impact of flax fibre content and preform architecture on the kinetics of degradation has been previously explored [5,6]. Moreover, mechanical properties of non-woven flax-PLA composite materials have been monitored during garden composting, showing a sudden decrease in tensile properties, but thereafter preservation of residual mechanical performance (*vis.* strength) at around 50% of initial strength even after six months in compost [2].

To our best knowledge, no existing studies focus specifically on the mechanical and structural degradation, and associated mechanisms, of the plant fibre constituents during the compost ageing process of biocomposites. Indeed, such investigations within a composite, at the microscale, are challenging due to difficulties in sample preparation and the limited number of sufficiently precise and resolved characterisation tools. The development of atomic force microscopy in Peak-Force Quantitative Nano-Mechanical mode (AFM-PF-QNM) [7] paves the way for original investigations with high precision and resolution allowing semi-quantitative indentation modulus mapping at the cell wall scale.

Here, following more than four months in a garden compost, a flax-PLA non-woven composite was examined using AFM-PF-QNM technology. The mechanical properties of the composite material, its constituent flax fibres, PLA matrix and fibre-matrix interfaces, were compared before and after composting. Specifically, cross-sections of flax cell walls were mechanically mapped in different areas of composite sections to understand flax fibre degradation mechanisms within a biocomposite. 

## 2. Materials and Methods

### 2.1. Materials

The industrial flax-PLA non-woven preform (100 ± 15 g/m^2^) was provided by Eco-Technilin (Yvetot, France). This light non-woven mat is made with a needle-punching line and extracted before the napping step, prior to consolidation. PLA 3001 D from NatureWorks was used to produce films with a thickness varying from 50 to 100 µm.

### 2.2. Composite Manufacturing and Composting Stage

Composite plates of 2 mm thickness were manufactured by a film stacking process and using thermo-compression, as described in a previous work [2]. Fibre volume fraction of 30 ± 1% and porosity level lower than 2% was achieved [2]. Dog-bone shaped samples drawn from ISO 527 were cut from these plates by a milling machine; due to the milling process, the edges of the specimen were not protected and therefore the flax fibre section was accessible (see Figure 1). Specimens were then aged in a garden compost made of green and brown plant waste [2]. Samples were buried at a depth between 20 and 40 cm; sampling was conducted after 125 days in the compost.

### 2.3. Composite Tensile Test 

Static tensile tests were performed on an Instron 5500R machine (Instron, Norwood, MA, USA) where displacement was recorded by an EIR LE-05 laser extensometer. The gauge length was 25 mm, and the displacement speed was set at 1 mm/min. At least 5 valid experiments points (e.g., ignoring grip failures) were used for statistical analysis. The tangent modulus was calculated in the strain range of 0.02% to 0.15%.

### 2.4. Sample Preparation for AFM Study 

One composite sample was cut from a tensile specimen (Figure 1c–e) after 125 days of composting. The sampling region (area and orientation; Figure 1e,f) was selected to include both ‘exposed’ flax fibres (near the composite edge/surface, close to the compost) and ‘protected’ flax fibres (in the core of the composite sample). The sample was embedded in agar resin (epoxy resin agar low viscosity resin (LV); Agar Scientific, Stansted, UK) and prepared for nanoindentation and microscopy as described in [7] (Figure 1g). Before micromechanical analysis, the sample was observed under an Olympus AX70 optical microscope equipped with a 5× (N.A. 0.15) BD objective to create a mosaic of the full surface (Figure 1h). 

### 2.5. AFM Investigations

A MultiMode 8 AFM instrument (Bruker, Billerica, MA, USA) was used in PF-QNM imaging mode. This mode is based on the recording of force–distance curves at a high rate (2 kHz) for a limited maximum load (200 nN here), and thus limited indentation depth (of the order of a nanometre here), while the tip scans the surface of the sample, thus allowing us to make maps. The indentation modulus is derived from the unloading part of the force–distance curve using an appropriate contact model. We used a DMT model here, which corresponds to the Hertz contact model (small indentation depth compared to the tip apex radius) modified to take into account the adhesion force (mainly due to water capillarity in our case) between the tip and the sample surface [8]. The indentation modulus obtained is similar to that obtained by nanoindentation measurements, but with the required resolution to study mechanical gradient within cell wall layers [9,10]. A RTESPA-525 (Bruker, https://www.brukerafmprobes.com/p-3915-rtespa-525.aspx, accessed on 13 March 2021) silicon probe with a spherical tip apex was used here. Its spring constant (between 136 and 177 N/m) was calibrated using the Sader method (https://sadermethod.org/, accessed on 13 March 2021) and the tip radius adjusted between 20 and 80 nm on a reference sample of similar indentation modulus as those of the fibre cell wall. The image resolution was 512 × 512, 384 × 384 or 256 × 256 pixels depending on the aim of the image captured. The peak force amplitude was set at 30 nm for PLA measurements and between 50 and 100 nm for fibres, depending on the region investigated. At least three images were used to calculate the average moduli and roughness of PLA. The root mean square (RMS) roughness was calculated after a flatten treatment (order 1) to remove the tilt of the sample beneath the tip. 

### 2.6. SEM Analysis 

Two different samples were prepared for SEM analysis: (i) a composite sample, cryo-fractured in nitrogen, for interface observation and also (ii) the sample prepared for AFML and described in Figure 1 was SEM observed after AFM investigations. All samples were gold-sputtered using a Scancoat6 from Edward. Then, these samples were observed under a JEOL SEM (JSM- IT500HRSEM) at an acceleration voltage of 3 kV.

## 3. Results and Discussion

### 3.1. Evolution of Overall Composite Microstructure with Composting Stage 

After spending 125 days in a garden compost, flax/PLA non-woven composites have undergone notable microstructural and mechanical property changes. As the interface region is a critical zone for stress transfer, the strength of aged composites was harshly impacted with a decrease of 51 ± 4% after composting. The degradation of the composite impacts not only its strength but also Young’s modulus, which decreased by 66 ± 7% (Figure 2a).

As is visible in Figure 2b,c, the degradation of the interfacial region, induced by a loss in the flax/PLA cohesion after ageing, plays an unarguable role in this drop in mechanical properties. This result is confirmed by previous results, for which a weight loss of 5.75 ± 0.57% was recorded [2] and the presence of internal porosity was observed. Looking at the position and geometry of this porosity, they appear to be mainly located at the interface between flax fibres and PLA. It is, therefore, relevant to also question the evolution in mechanical properties and ultrastructure of the flax fibres during this degradation, as they are largely responsible for the stiffness of the composite. 

### 3.2. Morphology of the Composite Section after Composting 

Figure 3 presents a view of the investigated face (Figure 3a), as well as SEM images of flax fibre elements (Figure 3b–d) and PLA matrix (Figure 3e–g) at specific locations of the sample. 

The sample was in direct contact with the compost at three faces (Figure 1a): face A and the two lateral faces B. Face A presented the worst conditions, as this face was milled during the specimen preparation resulting in a large number of fibre sections, devoid of PLA encapsulation, that are in direct contact with the compost. In contrast, as the upper and lower faces B of the specimens were in contact with platens during thermo-compression specimen manufacture, the surface and fibres would be well-permeated with PLA. Thus, the porous face A is more likely to promote the physical conditions necessary for the entry and proliferation of microorganisms [11] and, in this specific case, broken flax fibres provide an optimal access point and molecular resources for microorganisms. After 125 days in compost, degradation of PLA-flax fibre interface is noticed (Figure 3d), even in the core of the composite, as in Figure 2c; this creates voids at the fibre–matrix interfaces, probably induced by moisture sorption and desorption of flax fibres during ageing combined with the low deformation capacity of PLA. These interfacial cavities may further facilitate the penetration of microorganisms through “channels” to the core of the sample.

In Figure 3a, one can notice that fibres closest to the compost exposed face have a black colour in contrast to that of the fibres located in the core of the material, which are less accessible. Three stages of fibre degradation were identified: (i) mostly black fibres at an advanced stage of degradation that often result in large empty cavities, and in some cases with a cell wall residue in the periphery (Figure 3b); (ii) partially degraded cell walls characterised by a grey colour often associated with channels and pits (Figure 3c); and (iii) apparently intact fibres still aggregated in bundles with a clear colour (Figure 3d). Following this observation, for the purpose of discussions in this paper, we divided the composite sample into three specific regions: edge, middle and core (Figure 3a). 

Visually, at the SEM microscale, the PLA matrix is relatively less damaged in all three regions, though one finds an increase in porosity from the core to the edge (Figure 3e–g), with porosity size also increasing the closer they are to the edge. Development of progressive pores has already been documented for PLA, in case of immersion in a degradation medium (i.e., pH 7.4, 37 °C phosphate buffer) due to solubilisation of oligomers during 15 weeks, which is consistent with the duration of our composting stage [12]; it was evidenced that conjugated decrease of PLA molecular weight and water sorption were the key factors for this morphological degradation.

### 3.3. Assessment of the Flax Fibre Degradation Using AFM 

In literature, bacteria responsible for wood and plant fibre degradation are divided into tunnelling, erosion and cavitation bacteria based on the characteristic pattern they create during their attack [11,13,14]. The tolerance limits of bacteria, especially dependent on temperature, oxygen and high values of water activity (water molecules not chemically bonded to the material) [11], can make their proliferation difficult. However, in compost and in burial environments, optimal conditions allow bacteria to dominate over fungi [11]. An exception are the soft-rot fungi, particularly tolerant to these environmental conditions, which can coexist with bacteria, sometimes even degrading the same wood cell wall [15]. In general, it is possible to distinguish between fibre decay caused by fungi or bacteria, but sometimes their marks of degradation are similar, especially at an advanced stage of the attack, as demonstrated in archaeological wood in burial conditions [16]. 

The three stages of fibre degradation previously identified (Figure 3b–d) were successively investigated by AFM, for which results are shown in Figure 4. 

After 125 days in compost, in the sample core, some fibres were found to be still intact (Figure 4b,e,h) and with an indentation modulus around 18–23 GPa, in line with indentation moduli generally found for flax fibres recorded in other papers by AFM or nanoindentation [7]. Nevertheless, one can notice a beginning of degradation in these core fibres, with a low indentation modulus area around the lumen (Figure 4h) inducing a bimodal distribution of indentation modulus (Figure 4i) with two main peaks around 23 and 10 GPa. A progressive degradation, with first an attack of more labile polymers such as hemicellulose and pectins can be hypothesised. It should be considered that the section investigated in this work represents only a slice of the whole sample. Fibres and fibre bundles at different planes can be more or less degraded than the one investigated, as was demonstrated by Björdal et al. [17] thanks to tomographic analysis. In general, degraded regions were found at the edge of the fibres (rarely), randomly distributed in the secondary layers (typical in soft-rot decay, type I) [13] as shown in Figure 5d, or more often starting from the lumen (Figure 3c and Figure 4h). 

Fibres investigated in the middle region of the sample show more pronounced decay (Figure 3c and Figure 4g) than those in the core (Figure 3d), with an increase of the degraded area from the lumen region. More precisely, mechanical properties of the gelatinous G layers decreased considerably (around 5–7 GPa) in the degraded regions (Figure 4g,i). However, a part of the secondary wall still appeared intact with stiffness around 18 GPa (Figure 4i). The third step of fibre decay, especially in the edge of the biocomposite, consists in the complete degradation of the cell wall fragments, until only one cavity remains. Despite the advanced state of degradation of the whole cell, S1 and the last parts of the G layer, seemed to be partially preserved (Figure 4f) at the periphery of the fibre (Figure 5f). When fibre located at the extreme edge of the sample is considered (Figure 4a), cell walls are fully degraded. This loss of fibre material is mainly responsible for the increase in porosity of the biocomposite and its consequent loss of weight [2]. 

Thus, the described degradation mechanisms, from the lumen of the fibre, is a typical behaviour for tunnelling and erosion bacteria [14] but several fungi can also grow into the lumen (Figure 5e,f) and successively intrude into the rest of the fibre [18]. The progressive decrease in indentation modulus observed here, from the lumen to fibre periphery, is linked to the severity of the biological attack. It confirms also the predominant role of the lumen in plant fibre degradation; it enables the transport and propagation of water and microorganisms. Moreover, flax fibres are cellulose- and hemicellulose-rich and these components are high value resources for chemo-organotrophic microorganisms. Degradation is also promoted by their low lignin content, as lignin is known to inhibit the attack of fungi and bacteria. In Figure 4g, one can note that fibres investigated in the middle of the biocomposite lose their mechanical properties following the fibre shape, confirming that the gelatinous layers of flax, rich in cellulose and poor in xylan and lignin [19,20] are easily degraded; in the present case, cell wall degradation is probably initiated by structural hemicelluloses, less recalcitrant than crystalline cellulose.

### 3.4. Degradation of PLA

Contrary to flax fibres, PLA exhibits a low difference in indentation moduli between edge, middle and core regions (Figure 6e–g) with average values ranging from 4.7 ± 0.2 GPa to 5.2 ± 0.2 GPa. In contrast, a strong decrease in roughness, probably induced by the specific behaviour of the PLA during the cutting stage, from the core to the edge is evidenced with a ratio of 4.2 between core and edge mean values. 

The topography observed in the core region is comparable to that of PLA investigated in other studies [21,22], where the polymer appears as a complex matrix of fibrils, aggregated together. In our case, the polymer formed filaments around 150–400 nm in length (Figure 6d). During the first days of composting, the temperature under the compost can reach 60 °C [2] and this can cause a possible evolution of post-process local residual stresses as well as re-arrangement and recrystallisation of PLA during composting [2] that is visible in the core region (Figure 6d), with higher roughness, and disappears at the edge where the degradation is more severe (Figure 6b,c). This supports the results in Xu et al. [21], where the authors observed a similar change in the surface topography of PLA film subjected to hydrolysis after 60 and 90 days at 60 °C. In other research, an increase in crystallisation and mineralisation were noted after biodegradation and weathering processes in PLA/plant fibre composites [23,24,25] that correlate with this phenomenon of re-crystallisation. This increase in crystallisation may induce a minor improvement in the indentation modulus as evidenced in our sample (Figure 6), even though the change in roughness may interfere with the measurement of the indentation modulus.

In our case, the morphological change observed by AFM and decrease in RMS indicate an evolution in the organisation of PLA in the edge area in comparison to the core region, suggesting a role of microorganisms in secreting enzymes that degrade the polymers in oligomers, dimers and monomers being absorbable as nutrient [26,27]. Pits present in the PLA matrix, as observed by SEM (Figure 3e,f) or AFM (Figure 6c), are additional proof of this structural evolution, but it was not possible to attribute them to mechanical degradation caused by fungi or other types of ageing mechanisms such as self-catalysis hydrolysis [28].

A list of fungi and bacteria capable of degrading PLA are reported in [29] but, in Suyama et al. [30], PLA has been shown to be particularly resistant to biodegradation compared to other plastics. These authors also reported that only up to 0.04% of bacterial colonies present in soil have the capacity to degrade PLA. Despite its low sensitivity to biodegradation, it is interesting to note that, in the literature, PLA [27,31] and bast fibres [18,32,33] have some bacteria, such as *Bacillus brevis, Bacillus licheniformis, Paenibacillus amylolyticus*, and fungi, such as *Trichoderma viride* and *Fusarium moniliforme*, in common that are responsible of their biodegradation, but also used (or isolated) during the retting process for fibre extraction.

## 4. Conclusions

Here, we studied the compost ageing of PLA-flax non-woven composite materials. After a period of over four months in a garden compost, flax fibres within a PLA-flax composite undergo drastic degradation with a significant decrease in indentation modulus and progressive increase in cavities from the inner part, the lumen, to the periphery of the fibre. It was shown that the lumen is a preferential degradation channel, with cell wall degradation occurring through well-known tunnelling and erosion effects. In contrast, the indentation modulus of the PLA matrix is not particularly affected, but a significant evolution in PLA morphology and change in roughness, from the core to the edge of the sample, are highlighted. These phenomena are conjugated with the development of pits, induced by water ingress and microorganism attacks.

## Figures and Tables

**Figure 1 polymers-13-02225-f001:**
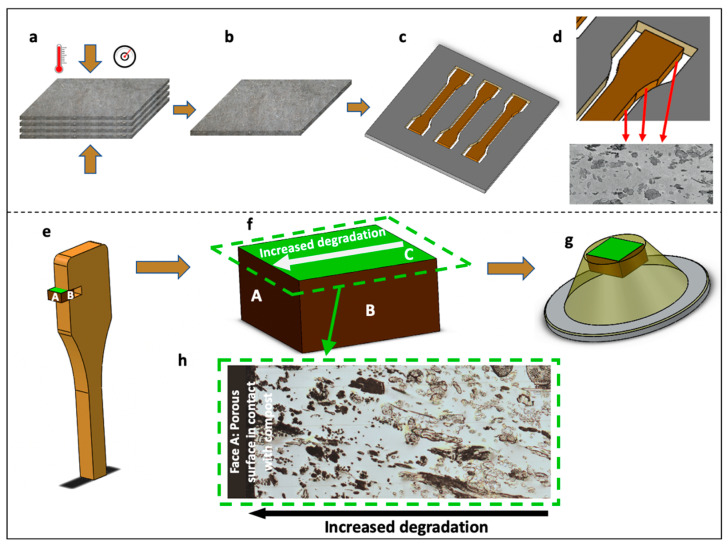
Manufacturing of the composite plates: (**a**) stacking of PLA-flax non-woven layers for hot compression; (**b**) is the obtained composite plate; (**c**) is the plate after the specimen cutting process; and (**d**) focuses on the structure of the edge of the specimen with apparent flax fibre cross-sections after cutting; (**e**) is a schema of the dog-bone sample indicating the region of sampling. The green face C (**f**,**g**) was specifically investigated in this paper; the extracted sample was embedded in agar low viscosity resin and glued on an AFM sample holder (**g**). Face A represents the most porous ‘exposed’ face (**h**), in contact with the compost, as the dog-bone sample was cut by a milling machine, whereas faces B are less porous external faces in contact with thermo-compression plates, and the green face C was the one investigated by optical, electronic and atomic force microscopy.

**Figure 2 polymers-13-02225-f002:**
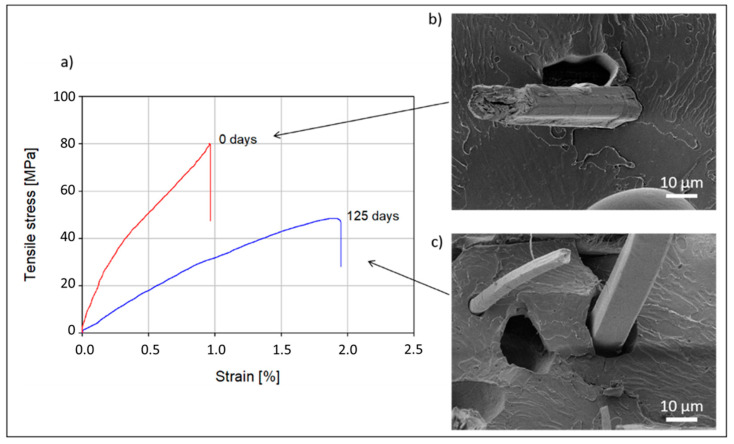
(**a**) Mechanical property evolution of a non-woven flax/PLA composite after spending 125 days in a garden compost; (**b**,**c**) SEM micrographs of nitrogen cryo-fractured samples, showing the interface between flax fibre and PLA before (**b**) and after (**c**) compost ageing.

**Figure 3 polymers-13-02225-f003:**
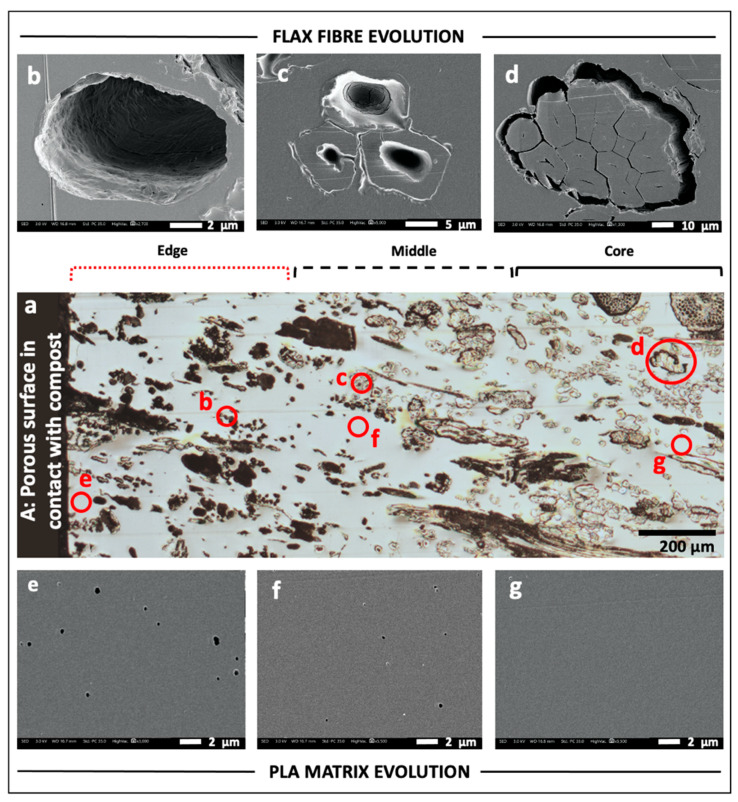
(**a**) Optical micrograph of the investigated face C (see Figure 1), which is divided into edge, middle and core areas following visual observation of fibre degradation; (**b**–**d**) SEM images of fibres collected in edge, middle and core, respectively. The complete degradation of the fibres leaves phantom cavities with the same shape of the degraded fibre (**b**). Different stages of degradation can be recognised in (**c**). In (**d**), fibres in core appear intact; (**e**–**f**) SEM images of PLA matrix investigated in edge, middle and core, respectively. Some porosities visible in edge (**e**) and middle (**f**), probably due to degradation, are absent in core (**g**). For each SEM image, the corresponding investigation area is indicated with red circles in (**a**).

**Figure 4 polymers-13-02225-f004:**
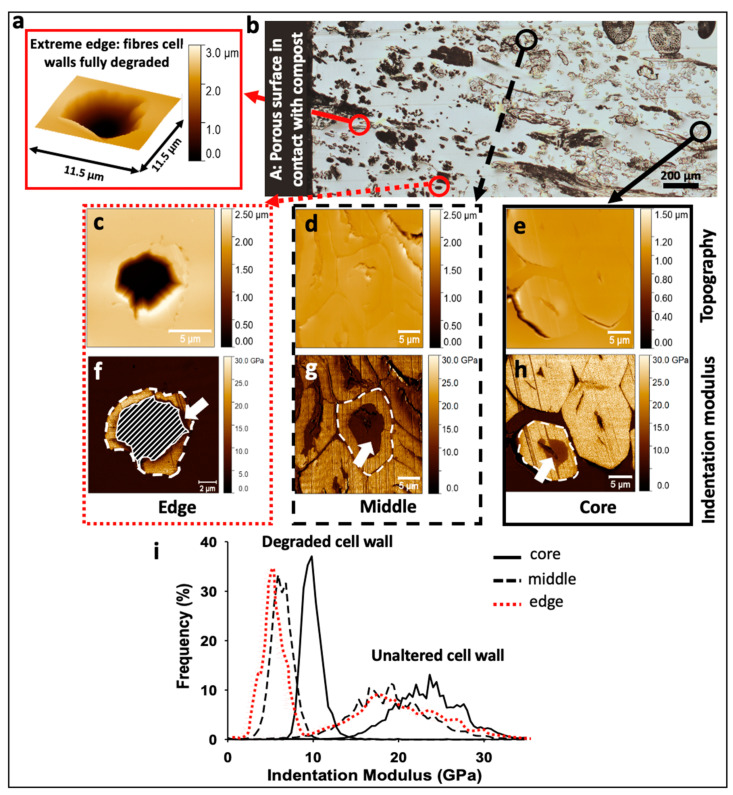
(**a**) 3D AFM topographic image of a fully degraded fibre, located at the extreme edge of the sample; (**b**) face C investigated through AFM-PF-QNM; black and red circles indicate the fibres selected for AFM measurements; (**c**–**e**) are AFM topographic images of edge, middle and core areas, respectively, and (**f**–**h**) the corresponding indentation modulus maps. (**i**) shows the indentation modulus distribution for each investigated region, indicated in white dotted lines in (**f**–**h**). In (**f**), no data were recorded in the hatched area.

**Figure 5 polymers-13-02225-f005:**
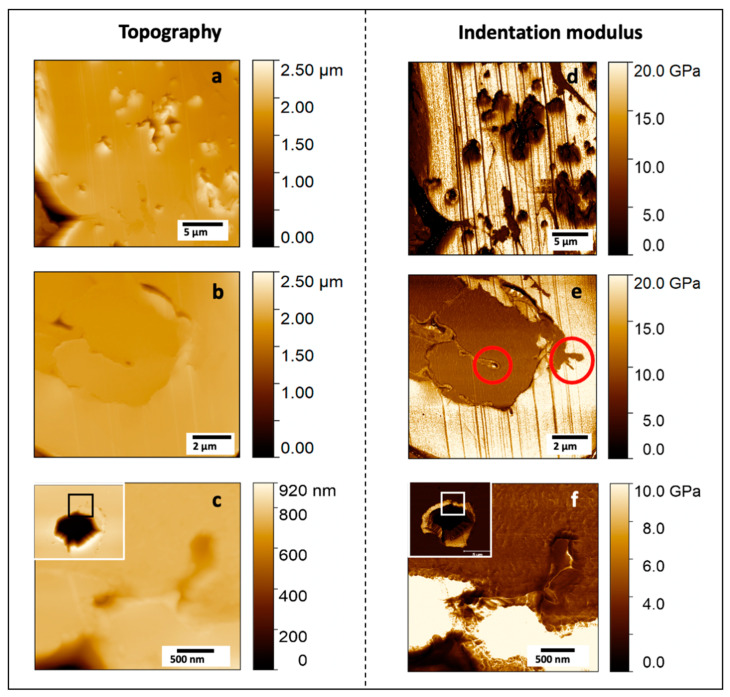
Surface topography (**a**–**c**) and indentation modulus mapping (**d**–**f**) of specific degraded areas. (**a**,**d**) are details of some fibres randomly degraded in G layers as typical degradation of soft-rot fungi; (**b**,**e**) focus on some typical degradation marks found in flax fibres, and red circles highlight the presence of bacteria and a probable fungal mycelium; (**c**,**f**) focus on a tunnel between inner and outer layers of the cell with a fungal mycelium that has grown in the exterior of the fibre and macrofibrils of the cell that are fragmented in packets with a low indentation modulus.

**Figure 6 polymers-13-02225-f006:**
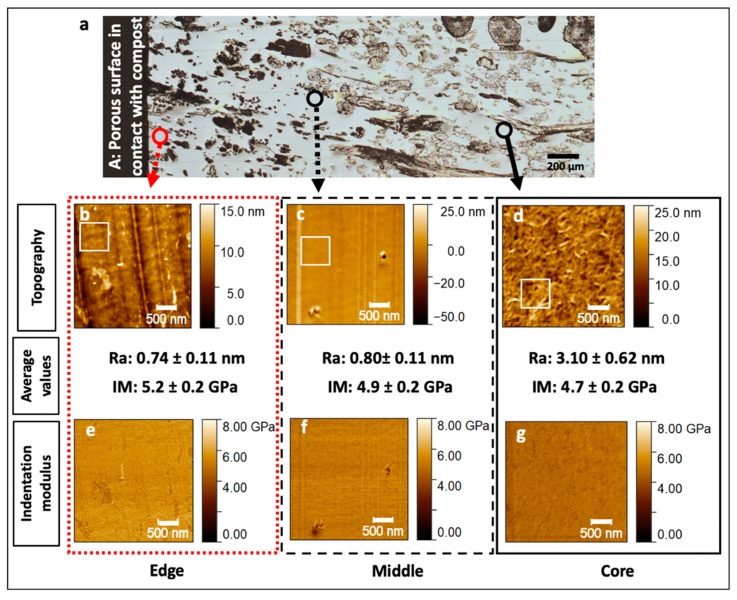
Maps of PLA topography and indentation modulus in the edge (**b**,**e**), middle (**c**,**f**) and core regions (**d**,**g**), respectively. Red and black circles on the investigated composite section (**a**) represent the areas examined through AFM. Average values of roughness (Ra) and indentation modulus (IM) are given. Average roughness was calculated through topography acquisition in the areas delimited by white squares (55 × 55 px) and average modulus was calculated in the whole image avoiding the defects when possible (around 65,500 force curves); at least three images were used to calculate the mean roughness and modulus in each region.

## Data Availability

Data supporting the findings of this study are available on simple request from the corresponding author.

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
