# Peer review of "Investigations by AFM of Ageing Mechanisms in PLA-Flax Fibre Composites during Garden Composting"

_polymers, 2021, doi:10.3390/polym13142225_

Round 1
Reviewer 1 Report
- The authors should introduce why they chose flax-PLA composites and their prospective applications in industry as non-experts will be curious as to why the authors chose this material.
- The authors should also introduce peak-force quantitative nanoindentation and how it is useful for analyzing fiber materials compared to conventional nanoindentation methods.
- How does the measured modulus of fibers differ with the use of peak-force quantitative nanoindentation compared to normal nanoindenation?
- What was the tip material and geometry in their AFM peak-force quantitative nanoindentation experiments?
- The materials section last sentence, the units are ambiguous.
- Why was garden composting method used compared to other compositing methods?
Author Response
Dear Dr. Thumawadee Wongwirat,
We kindly thank the referees who have taken the time to give us meaningful feedback on our work, which has helped us improve the quality of our manuscript. We hope they will find satisfaction in our responses (outlined below).
The various changes we have made are in red font in the new version of the manuscript.
Looking forward to hearing from you,
With best regards,
Alain Bourmaud
Reviewer 1
- The authors should introduce why they chose flax-PLA composites and their prospective applications in industry as non-experts will be curious as to why the authors chose this material.
The reviewer is right, we need to better introduce the overall industrial or societal concerns linked to this work. Today, flax-PP non-woven is a widespread material in the automotive industry, the use of a degradable matrix such as PLA is of great interest in other markets, especially for advertising applications. Industrial developments are in progress in this sector to design products and panels with short life and potential degradation as an end-of-life scenario. The time life and degradation of the products is of interest to understand the degradation mechanisms occurring during the ageing of the parts. Abstract and introduction have been modified to better highlight the objectives of the work.
- The authors should also introduce peak-force quantitative nanoindentation and how it is useful for analyzing fiber materials compared to conventional nanoindentation methods.
In AFM, a laser beam is focused on a tip mounted on a cantilever and the laser reflection is recorded by a photodetector. The deflection of the cantilever, due to the tip-sample interaction force, and the relative displacement of the sample produces a force-distance curve when the tip indents the sample surface, similar to the loading/unloading curve obtained by nanoindentation. Peak force quantitative nanomechanical property mapping (PF-QNM) mode is based on the recording of force-distance curves at a high rate (2 kHz) for a limited maximum load (200 nN here), and thus limited indentation depth (of the order of a nanometre here), while the tip scans the surface of the sample. This provides a map of indentation moduli, typically 512×512 or 1024×1024 points, in a few minutes to tens of minutes for a surface of several square micrometres.
Nanoindentation is a widespread and quantitative method for mechanical characterization at the micrometre scale and is now well developed for studying plant fibres at the whole cell wall scale (Bourmaud and Baley, Compos. Part B, 2012, doi:10.1016/j.compositesb.2012.04.050; Eder et al., Wood Sci. Technol., 2013, doi: 10.1007/s00226-012-0515-6). However, it is less well suited for studying gradients within the cell wall layers, due to the minimum indent size and required distance between indents (Sudharshan Phania and Oliver, Mater. Des., 2019, doi: 10.1016/j.matdes.2018.107563): few nm between each indentation measurements in AFM against a few µm for conventional nanoindentation, while the thickness of the cell wall layers varies from a few hundred nanometers to a few micrometers in general. Even if AFM measurements are less quantitative than nanoindentation, they are thus particularly interesting for evidencing local phenomena or gradients or properties within thin layers, such as differences into plant cell wall maturity or local properties of specific constitutive layers (Arnould and Arinero, Compos. Part A, 2015, doi:10.1016/j.compositesa.2015.03.026; Arnould et al, Ind. Crop. Prod., 2017, doi:10.1016/j.indcrop.2016.12.020; Melleli et al, Molecules, 2020, doi:10.3390/molecules25030632).
A shorter version of these explanations and comments has been added to the materials and methods part (2.6)
- How does the measured modulus of fibers differ with the use of peak-force quantitative nanoindentation compared to normal nanoindenation?
Nanoindentation (‘‘deep’’ penetration depth of some hundreds of nanometres using a diamond Berkovich 3-sided pyramid indenter) and AFM PF-QNM (‘‘small’’ penetration depth of some nanometres with an almost spherical tip apex) measure the same elastic property, i.e. the indentation modulus, with the slope of the unloading part of the indentation curve, but not at the same scale and with strong plasticity in the first case (which makes it possible to measure the hardness of the material). Moreover, the surface preparation process (cutting here) inevitably leads to near-surface modification with more or less significant impact on the mechanical properties (e.g., creating nano-cracks or molecular reorientation in the wake of the knife). As AFM measurements are done at lower scales than nanoindentation, they’re more sensitive to the sample surface properties and its nano-roughness. This nano-roughness can lead to an overestimation of the real contact area and yields an estimated indentation modulus lower than the real one for a given contact stiffness (Stan and Cook, Nanotechnology, 2008, doi:10.1088/0957-4484/19/23/235701). Finally, AFM measurements are more sensitive to nano-scale effects like adhesion or interaction forces. Thus, the contact model used to determine the indentation modulus from the contact stiffness must take these nano-effects into account, unlike those used in nanoindentation. The appropriate model depends on the adhesion and/or interaction forces and the contact stiffness (Johnson and Greenwood, J. Coll. Interf. Sci., 1997; Barthel, J. Phys. D, 2008, doi: 10.1088/0022-3727/41/16/163001). It depends also on the ratio between the indentation depth and the tip apex radius (e.g., Hertz vs Sneddon contact model). We use a DMT model here, which corresponds to the Hertz contact model (small indentation depth compared to the tip apex radius) modified to take into account the adhesion force (mainly due to water capillarity in our case) between the tip and the sample surface. Our experience in characterising plant fibres using PF-QNM and nanoindentation shows that both methods give similar indentation moduli with our sample surface preparation and the settings used during the experiments (Arnould et al, Ind. Crop. Prod., 2017, doi:10.1016/j.indcrop.2016.12.020).
We just mentioned in the materials and methods part (2.6) that AFM and nanoindentation measurements give similar results.
- What was the tip material and geometry in their AFM peak-force quantitative nanoindentation experiments?
As mentioned in the Materials and Methods section, AFM probes used here come from Bruker with the reference RTESPA-525: etched silicon probe with a close to conical shape tip (10-15 µm height) with a spherical apex having an initial radius of ≈10 nm (calibrated between 20 and 80 nm during the experiments). All specifications are given on Bruker’s website: https://www.brukerafmprobes.com/p-3915-rtespa-525.aspx
We have completed the information on the probe accordingly (2.6).
- The materials section last sentence, the units are ambiguous.
Indeed, this sentence wasn’t clear enough. A clearer formulation is employed to explain that the thickness of the film varies from 50 to 100 µm. line 78
- Why was garden composting method used compared to other compositing methods?
The main objective of the present study is to explore the material behaviour in a harsh but not destructive environment to consider extreme use conditions; for this reason, industrial composting was not considered. This paper follows another one on the degradation (Pantaloni et al., Polymer Degradation and Stability, 2020, https://doi.org/10.1016/j.polymdegradstab.2020.109166), in the same conditions, of a range of biopolymers-flax non-woven composite materials. The present one is an additional comprehensive work conducted at the fibre scale, to finely explore and understand the fibre degradation mechanisms. Consequently, we consider it is important to keep the same ageing conditions.
Reviewer 2 Report
Atomic force microscopy was used to investigate PLA-flax fiber composited after composting for 125 days. This manuscript was well-written. However, other testing like DSC, TGA and micro CT should be done to more understand effects of composting.
Author Response
Dear Dr. Thumawadee Wongwirat,
We kindly thank the referees who have taken the time to give us meaningful feedback on our work, which has helped us improve the quality of our manuscript. We hope they will find satisfaction in our responses (outlined below).
The various changes we have made are in red font in the new version of the manuscript.
Looking forward to hearing from you,
With best regards,
Alain Bourmaud
Reviewer 2
- Atomic force microscopy was used to investigate PLA-flax fiber composited after composting for 125 days. This manuscript was well-written. However, other testing like DSC, TGA and micro CT should be done to more understand effects of composting.
The reviewer is right, it would have been interesting to supplement this work by thermal analysis of PLA behaviour, through DSC or TGA analysis. In our case, ageing is monitored on one sample, with local variations of fibre volume fraction and different phenomena occurs in very limited regions of the specimen; consequently, conventional DSC or TGA is not possible on this kind of sample. One way is to use micro-DSC which is a promising method to investigate specific samples; today, we do not have access to this kind of equipment but the option will be considered in future work to deeply analyze the specific local thermal behaviour of PLA. Moreover, the first goal of this work is to explore the fibre and not PLA behaviour, even if this latter is of interest.
Micro CT is also a powerful method to explore the damages or porosity within a sample. We have in-progress experiments on this topic on similar samples, after specific ageing in different environmental conditions. A new work, specifically dedicated to this topic will be submitted soon.
Round 2
Reviewer 2 Report
The manuscript is in the perfect from for publication.